# The influencing factors of biomedical R&D cooperation in three major urban agglomerations of China based on cooperative patents

**Guojun Sun‡, Shaoya Zhang‡, Lan Xu, Xiaoying Zhou, Shuaijun Wu, Dong Zuo-jun⊕\***

Zhejiang University of Technology, Hang Zhou, China

‡ GS and SZ are contributed equally to this work and should be considered as co-first authors.
\* jzd1970@zjut.edu.cn

**Data Availability Statement:** All relevant data are within the manuscript and its Supporting

## Abstract

Due to the particularity of biomedical industry, it has become necessary for biomedical enterprises to seek innovative research and development (R&D) cooperation to maintain advanced technologies and products in multiple fields. Under such circumstance, the biomedical industry has gradually formed a certain cluster to promote the development of the industry. So far, the biomedical industry cluster has formed in China, mainly within the Yangtze River Delta, Pearl River Delta, and Beijing-Tianjin-Hebei three urban agglomerations. Within the industrial clusters, the frequency of innovation cooperation among enterprises, universities, research institutions, and other relevant organizations in the biomedical area is high, and the capacity for innovation cooperation is strong as well. This paper used the representative cross-section data of cooperative patents from the medical science and technology patent database of China National Knowledge Infrastructure (CNKI), researching the R&D cooperation within the three major urban agglomerations in China from 2008 to 2016 (Yangtze River Delta Urban Agglomeration, Pearl River Delta Urban Agglomeration, Beijing-Tianjin-Hebei Urban Agglomeration) on total 36 cities' spatial pattern characteristics of biomedical cooperation and the influencing factors. The spatial interaction model was used to study the spatial, economic, political, and R&D influencing factors of cross-city cooperation. The degree of aggregation showed that cross-city R&D cooperation mainly occurred in well-developed and central cities of urban agglomerations. Econometric results revealed that spatial, economic, political, and R&D bias factors did have a significant impact on the frequency of biomedical R&D cooperation across cities.

## 1. Introduction

Biomedicine is a high-tech, high-risk, high-input, technology-intensive, and knowledge-intensive industry, with great importance of intellectual property protection [1]. Patents are the key output forms of biomedical technologies, and also the important way to protect biomedical

Information files. Also available at https://doi.org/10.17026/dans-xw3-eqzx.

**Funding:** This research was funded by Research on the Construction Path of New Drug Creation and Innovation Consortium in the Field of Life and Health in Zhejiang Province, project number: 2022C25007.

**Competing interests:** The authors declare no conflict of interest.

intellectual properties. Due to the characteristics of biomedical industry, it is necessary to take advantage of the cluster effect of multi-party cooperation to achieve competitiveness, which is mainly reflected from joint-applications for patents with external organizations that easily forms a certain scale of cooperative patent application network [2]. Cooperative patents are the most direct and effective index to evaluate R&D cooperation and output.

Urban agglomeration is the product of industrialization and urbanization at an advanced stage and the important foundation of regional economic spatial patterns. In China, the competition pattern of the biomedical industry has taken shape, with the industry mainly concentrated in the three major urban agglomerations of the Yangtze River Delta, Beijing-Tianjin-Hebei region and Pearl River Delta [3]. The three urban agglomerations are always in the leading position on various policies given by the state to the biomedical industry in China. Among them, the Yangtze River Delta urban agglomeration has superior geographical conditions, excellent natural endowments, solid economic foundations, relatively mature system, the largest number of multinational biomedical enterprises in China, the highest level of innovation capabilities, good financing environment and active international exchanges of regional biological industry. In addition, different from other urban agglomerations in China, the Yangtze River Delta urban agglomeration consists of one major city together with multiple secondary cities, equipped with the "Zhangjiang" medicine valley positioned in Shanghai, the leading city in pharmaceutical industry of China, Taishan pharmaceutical industry city and Wuxi life science and technology industrial park [4], which belong to the aggregate development strategy and win by quality. The Beijing-Tianjin-Hebei urban agglomeration is China's "capital economic circle" and an urban agglomeration, with Beijing's core role as capital, radiating to surrounding cities so as to develop into a world-class urban agglomeration. It has rich human resources in biomedical field, highly complementary industrial chains and abundant clinical resources and educational resources [5]. The Pearl River Delta urban agglomeration is the leading region of China's reform and opening up and an important economic region in China. It plays an important role in promoting the national economic and social development in process of reform and opening up. Adjacent to Hong Kong and Macao, its pharmaceutical circulation system has been well developed, with strong external radiation capabilities, relatively active private capitals, a leading scale of biomedical equipment in China, and a relatively mature industrial system centered on innovative drug R&D, industrialization, formulation export and biomedical R&D outsourcing [6], belonging to a decentralized development strategy with quantity as its priority.

The selected three major urban agglomerations, including both coastal urban agglomerations and one central urban agglomeration, combining the actual conditions and distinctive characteristics of each urban agglomeration, represent three typical economic development models in China. The research on the influencing factors of R&D cooperation in these three types of urban agglomerations can bring corresponding enlightenment to the development of other urban agglomerations in China.

## 1.1 Literature review

Relevant literatures have been referenced on cooperative patents and R&D cooperation of pharmaceutical industry in the three major urban agglomerations. From the perspective of biomedical industry value chain,.PuRun etc, based on the perspective of industrial cluster, comprehensive analysis of the industrial cluster in the role of biological medicine industry innovation and development [6]. Production city journal pointed out that in the urban agglomeration "biological medicine industrial cluster" synergistic effect is more apparent [7]; Li jie, etc. Based on the theory of complex adaptive system model, analysis of biomedical

innovation clusters aggregation, nonlinear characteristics, flow, diversity and identification mechanism, internal model and block mechanism [8]; Xue Qian dimension of patents such as global and China biological medicine industry is analyzed, studied the technology branch of the research, development, advantage and put forward the industry-university-institute cooperation enterprises, major technological breakthrough and strengthen the countermeasures and Suggestions such as patent layout [9]. However, most of the above studies analyzed the cooperative evolution, motivation mechanism, performance, and influencing factors of the biomedical industry, while only a few studies quantitatively analyzed the factors influencing the inter-regional R&D cooperation [10–12], and even fewer empirical studies analyzed the influencing factors of pharmaceutical R&D cooperation with cities as nodes.

## 1.2 Hypothesis

The possible influencing factors for regional R & D cooperation are as follows: Firstly, due to the differences in natural environment, policy system, and industrial structure in various regions of China, regional economic developments are imbalanced in China [13, 14], the faster economic development, the stronger capabilities of regional scientific and technological innovation and the more cooperation. Secondly, geographical proximity between cities is an important factor to promote the spatial agglomeration and formation of urban scale radiation effect [15]. The planning and construction of a convenient inter-regional transportation network have gradually enhanced the positive impact on regional cooperation in scientific and technological innovation. Thirdly, the policy bias to promote R&D cooperation. Through empirical analysis of panel data from 30 provinces, autonomous regions, and municipalities in China from 2008 to 2016, He Baocheng [16] concluded that, for the provinces, autonomous regions, and municipalities in eastern and central regions, financial investment in science and technology had significantly promoted the transformation and generation of scientific and technological innovation achievements. In terms of science policy, the provincial government turned to protect local enterprises and research institutions, and universities to maximize the benefits of its own province [17]. Because of provincial protectionism, R&D collaboration between research institutes located in different provinces has encountered more obstacles. Fourthly, first-tier cities and provincial capitals are hubs of economy and politics [18], which is more advantageous for R&D cooperation.

Based on the above researches, the following hypotheses can be proposed: 1. Large economic differences reduce the possibility of R&D cooperation; 2. If one partner in the cooperation has the advantage of high-speed railway transportation, it has a positive impact on R&D cooperation; 3. Geographical distance affects R&D cooperation; 4. If the potential cooperation organizations are located in different provinces, the possibility of R&D cooperation is reduced; 5. If one partner is located at the provincial capital, it has a positive impact on R&D cooperation; 6. R&D cooperation can be promoted if one partner is located at a first-tier city or a central city; 7. The less difference in the development of pharmaceutical industry between two cities, the more positive impact on the R&D cooperation.

## 2. Materials and methods

### 2.1 Inclusion and exclusion criteria

The patent information comes from the medical and health science and technology database of CNKI, obtained from the patent data service platform of the State Intellectual Property Office.

**2.1.1 Inclusion criteria.** The cooperative patents jointly applied in all biomedical fields in three major urban agglomerations are included in the 2008–2016 CNKI medical and health

science and technology database. Extract the patent information of cooperative applications between organizations, including year, patent name, patent application number, patent applicant, address, and patent profile.

**2.1.2 Exclusion criteria.**   1.  Patents of natural persons as co-applicants;

2.  Patents jointly applied by foreign institutions in China.

## 2.2 Research design

The external variables of cross-city biomedical R&D cooperation were studied, and a spatial interaction model was proposed. This model has been previously applied to cross-city cooperation research [19–22]. Through the model, the influencing factors of the cross-city R&D cooperation of biomedicine were explored to verify the above hypotheses.

The spatial interaction model can be expressed as:

$$F_{ij} = O_i^{\alpha_1} D_j^{\alpha_2} \exp\left[\sum_{k=1}^{k} \beta_k S_{ij}^{(k)}\right] + \varepsilon_{ij}$$

i,j = 1,. . .,n

Where in $F_{ij}$ is the dependent variable, denoting the cooperation intensity between city i and city j, which is obtained by calculating the number of cooperation between city i and city j. $O_i$ and $D_j$ are the control variables for R&D cooperation scale, which denote the number of enterprises above the designated size in city i and city j respectively.$\alpha_1$ and $\alpha_2$ denote the unknown determination coefficients. $S_{ij}^{(k)}$ denotes K independent variables, means K factors affecting R & D cooperation between the two places; and $\beta_k$ (k = 1,. . ., K) is an unknown parameter to estimate the impact of various factors on R&D collaboration intensity; $\varepsilon_{ij}$ is the residual term of the model. In this paper, the research focuses on K = 7 independent variables. Based on the hypothetical mainline in this study, these variables can be grouped into the following three categories:

i.   Spatial variables: (1) $S_{ij}^{(1)}$ denotes the geographical distance calculated by spherical distance between city i and city j. (2) $S_{ij}^{(2)}$ is the dummy variable of high-speed rail. If there is a high-speed railway connection between city i and city j, the value is 1, otherwise is 0.

ii.  Economic and technological variables: (3) $S_{ij}^{(3)}$ is the economic difference between the two cities. By calculating the GDP difference between city i and city j in any one year of five-year; (4) $S_{ij}^{(4)}$ is the dummy variable of first-tier cities. If one or both of the cooperative partners are located at first-tier cities, the value is 1, otherwise is 0;

iii.  Political bias factors: (5) $S_{ij}^{(5)}$ is the dummy variable of different provinces. If city i and city j are in different provinces, the value is 1, otherwise is 0; (6) $S_{ij}^{(6)}$ is the dummy variable of the provincial capital city, if one or both city i and city j are provincial capital cities, the value is 1, otherwise is 0; (7) $S_{ij}^{(7)}$ is the dummy variable of the urban agglomeration center, if one of the two cities is the center of urban agglomeration (i.e. Beijing, Shanghai or Guangzhou), the value is 1, otherwise is 0.

iv.  R&D factor variables: (8) $S_{ij}^{(8)}$ is the dummy industry variable, obtained from the natural logarithm of the absolute value of biomedical manufacturing industrial output difference between the two cities. Due to the limitation of the nature and quantity of pharmaceutical patent cooperation, the data could be too discrete. The maximum likelihood estimation

method used by Poisson regression is usually applicable to non-normally distributed data. In this study, the observed cooperation intensity between cities presented Poisson distribution with a conditional mean. However, there were also many cities without cooperative relationships. Therefore, the number of 0 dependent variables was greater than the number of Poisson distribution hypotheses, and the conditional variance was greater than the conditional mean. The most critical difference between the negative binomial model and the Poisson regression model was the difference in conditional variance, by introducing additional parameter α into the negative binomial 5 type model, the problem of data dispersion could be solved.

## 2.3 Data collection and processing

Statistical analysis was performed on the information meeting the inclusion criteria.

**2.3.1 City selection.** The Yangtze River Delta urban agglomeration contains four administrative regions, including three provinces and one municipality directly under the central government. The Beijing-Tianjin-Hebei urban agglomeration consists of three administrative regions, including one province and two municipalities directly under the central government. The Pearl River Delta urban agglomeration covers only one province. According to China Statistical Yearbook in 2021, we selected 25 prefecture-level cities from the 4 administrative regions of Yangtze River Delta urban agglomeration, 9 prefecture-level cities from the 3 administrative regions of Beijing-Tianjin-Hebei urban agglomeration, and 12 prefecture-level cities from the 1 administrative region of Pearl River Delta urban agglomeration as samples (Select the city population base > 21.89 million).

**2.3.2 Build cross-city R&D cooperation matrix.** Firstly, we searched the patent data from 2008 to 2016 by using the patent database of CNKI within the range of medical and health technology, then further screened out at least two cooperative patents each from different R&D organizations (companies, universities, or research institutions). According to the statistics of pharmaceutical cooperative patents in the following periods(because each year's data is too little, in order to convenient statistics, data statistics once every two years): 2008–2010, 2011–2013, and 2014–2016, the three major urban agglomerations were respectively constructed to build the cross-city cooperation frequency matrix of 25×25 (Yangtze River Delta urban agglomeration), 9×9 (Beijing-Tianjin-Hebei urban agglomeration) and 12×12 (Pearl River Delta urban agglomeration). The number of cooperative R&D activities among organizations could be converted into the number of cooperation between cities, namely, into the number of R&D cooperation between city i and city j. If a patent was completed by organizations covering three different cities (city k, city m, and city n), there should be three kinds of patent cooperation: city k—city m, city k—city n, and city m—city n. The R&D cooperation data were recorded in the symmetric matrix, containing the cooperation patent links between cities. The resulting cooperation matrix (row i, column j) contained the cooperation intensity ($I = 1,. . ., n = 25/9/12; J = 1,. . ., n = 25/9/12$). The N×N matrix was designed symmetrically ($pij = pji$), and each value in the matrix should be the sum of cooperative relationships between city i and city j. This study mainly focused on cross-city cooperation, so we excluded the relationship of intra-city cooperation in the spatial interaction model.

Table 1 describes and lists the data sources of dependent variables and independent variables in the spatial interaction model, using Stata software for all-around analysis.

**2.3.3 Data processing.** Descriptive statistical analysis of pharmaceutical common patents by time periods; The top 5 cities of cross-city cooperation on joint patents by time periods; Results of the spatial interaction model by time periods (2008–2010, 2011–2013, 2014–2016) to verify the hypothesis that influencing factors are external variables. Data entry and

**Table 1. Variable description and data sources.**

| Variable type | Variable names | Description | Data source |
|---|---|---|---|
| Dependent variable | $F_{ij}$: R&D cooperation between city i and city j | Obtain numbers of cooperation among cities from captured co-patent data | China Patent Database |
| Independent variable | $O_i$: Primitive variables | Natural logarithm of the total number of enterprises in city i | www.drcnet.com.cn |
| | | | State Statistical Bureau |
| | $D_j$: Target variables | Natural logarithm of the total number of enterprises in city j | www.drcnet.com.cn |
| | | | State Statistical Bureau |
| Geographical factor variables | $S_{ij}^{(1)}$: Separation variables to measure geographical distance | Natural logarithm of the center spherical distance between city i and city j | Chinese Geographical |
| | | | Informational System Data |
| | $S_{ij}^{(2)}$: Dummy variable of high-speed railway | 1 = Cities i and j with high-speed railway connection; | China Railway Press |
| | | 0 = Cities i and j without high-speed railway connection. | |
| Economic and technological factor variables | $S_{ij}^{(3)}$: Separation variables to measure the economic disparity between city i and city j | Natural logarithm of the absolute value of GDP difference between cities i and j | www.drcnet.com.cn |
| | | | State Statistical Bureau |
| | $S_{ij}^{(4)}$: Dummy variable of first-tier cities | 1 = Either or both cities (i and j) are first-tier cities in China; | China City Statistical Yearbook |
| | | 0 = Neither city i nor city j is first-tier city in China | |
| Political bias factor variable | $S_{ij}^{(5)}$: Dummy variable of different provinces | 1 = Cities i and j in different provinces; | China City Statistical Yearbook |
| | | 0 = Cities i and j in the same province | |
| | $S_{ij}^{(6)}$: Dummy variable of the provincial capital | 1 = Either or both cities (i and j) are provincial capitals; | China City Statistical Yearbook |
| | | 0 = Neither city i nor city j is provincial capital | |
| | $S_{ij}^{(7)}$: Dummy variable of central city in the urban agglomeration | 1 = Either city i or city j is the central city in urban agglomeration; | China City Statistical Yearbook |
| | | 0 = Neither city i nor city j is the central city in urban agglomeration. | |
| R & D environment factor variable | $S_{ij}^{(8)}$: Dummy variable of industry | Natural logarithm of the absolute value of biomedical manufacturing industrial output difference between city i and city j. | www.drcnet.com.cn |

descriptive statistical analysis were performed by using the software Excel2010. Charts were made by writing in R language.

## 3. Results

### 3.1 Descriptive statistical results

Descriptive statistics of R&D cooperation among the three major urban agglomerations were obtained from the data of biomedical common patents based on three time periods, 2008–2010, 2011–2013, and 2014–2016. As shown in the table below.

**3.1.1 Yangtze river delta urban agglomeration.** Tables 2–4 lists the results of a descriptive statistical analysis of R&D cooperation in the Yangtze River Delta urban agglomeration.

**Table 2. Descriptive statistics of the Yangtze River Delta 2008–2010.**

| | ME | Sum | Mean | SD | Min | Max | Skewness | Kurtosis |
|---|---|---|---|---|---|---|---|---|
| All R&D cooperation | 63 | 854 | 13.5555 | 49.57 | 1 | 378 | 6.61 | 48.23 |
| Cross-city R&D cooperation | 47 | 247 | 5.2553 | 7.34 | 1 | 32 | 2.52 | 8.69 |
| Intra-provincial cross-city R&D cooperation | 23 | 82 | 3.5652 | 6.07 | 1 | 28 | 3.21 | 12.88 |
| Inter-provincial cross-city R&D cooperation | 24 | 165 | 6.875 | 8.18 | 1 | 32 | 2.16 | 6.70 |

**Table 3. Descriptive statistics of the Yangtze River Delta 2011–2013.**

|  | ME | Sum | Mean | SD | Min | Max | Skewness | Kurtosis |
|---|---|---|---|---|---|---|---|---|
| All R&D cooperation | 67 | 1135 | 16.9403 | 60.24 | 1 | 482 | 7.09 | 54.77 |
| Cross-city R&D cooperation | 49 | 333 | 6.7959 | 8.72 | 1 | 41 | 1.97 | 6.70 |
| Intra-provincial cross-city R&D cooperation | 25 | 133 | 5.32 | 6.77 | 1 | 26 | 1.92 | 5.75 |
| Inter-provincial cross-city R&D cooperation | 24 | 200 | 8.3333 | 10.30 | 1 | 41 | 1.69 | 5.32 |

**Table 4. Descriptive statistics of the Yangtze River Delta 2014–2016.**

|  | ME | Sum | Mean | SD | Min | Max | Skewness | Kurtosis |
|---|---|---|---|---|---|---|---|---|
| All R&D cooperation | 67 | 1436 | 21.4328 | 66.88 | 1 | 516 | 6.37 | 46.46 |
| Cross-city R&D cooperation | 48 | 449 | 9.3542 | 13.90 | 1 | 63 | 2.43 | 8.28 |
| Intra-provincial cross-city R&D cooperation | 23 | 216 | 9.3913 | 15.85 | 1 | 63 | 2.54 | 8.19 |
| Inter-provincial cross-city R&D cooperation | 25 | 233 | 9.32 | 12.17 | 1 | 48 | 2.03 | 6.57 |

From the data of the three time periods in the table, it can be seen that the R&D cooperation volume of the Yangtze River Delta urban agglomeration was rising, and the proportion of cross-city R&D cooperation was also rising. The total amount of intra-provincial cross-city R&D cooperation during the three time periods was lower than that of inter-provincial cross-city R&D cooperation during the same time periods, and as the time delay in the province of research and development of intercity cooperation are lower than the less degree of total provincial research and development of intercity cooperation. From 2008 to 2010, the total amount of intra-provincial cross-city R&D cooperation was only half of that of inter-provincial R&D cooperation, but from 2014 to 2016, the total amount of intra-provincial cross-city R&D cooperation was slightly different from that of inter-provincial R&D cooperation. Differences between intra-provincial and inter-provincial cooperation might indicate the influence of different provinces, the extent of this possibility would be verified in the spatial interaction model below.

### 3.1.2 Beijing-Tianjin-Hebei urban agglomeration

Tables 5–7 lists the results of a descriptive statistical analysis of R&D cooperation in the Beijing-Tianjin-Hebei urban agglomeration. As the coordinated development strategy of Beijing-

**Table 5. Descriptive statistics of Beijing-Tianjin-Hebei region 2008–2010.**

|  | ME | Sum | Mean | SD | Min | Max | Skewness | Kurtosis |
|---|---|---|---|---|---|---|---|---|
| All R&D cooperation | 8 | 558 | 69.75 | 132.59 | 1 | 372 | 1.74 | 4.49 |
| Cross-city R&D cooperation | 4 | 32 | 8 | 8.76 | 1 | 20 | 0.67 | 1.85 |
| Intra-provincial cross-city R&D cooperation | 1 | 2 | 2 | ---- | 2 | 2 | ---- | ---- |
| Inter-provincial cross-city R&D cooperation | 3 | 30 | 10 | 9.54 | 1 | 20 | 0.19 | 1.5 |

**Table 6. Descriptive statistics of Beijing-Tianjin-Hebei region 2011–2013.**

|  | ME | Sum | Mean | SD | Min | Max | Skewness | Kurtosis |
|---|---|---|---|---|---|---|---|---|
| All R&D cooperation | 11 | 538 | 48.9091 | 117.36 | 1 | 392 | 2.58 | 8.06 |
| Cross-city R&D cooperation | 8 | 36 | 4.5 | 6.70 | 1 | 20 | 1.81 | 4.75 |
| Intra-provincial cross-city R&D cooperation | 1 | 2 | 2 | ---- | 2 | 2 | ---- | ---- |
| Inter-provincial cross-city R&D cooperation | 7 | 34 | 4.8571 | 7.15 | 1 | 20 | 1.60 | 3.99 |

**Table 7. Descriptive statistics of Beijing-Tianjin-Hebei region 2014–2016.**

|  | ME | Sum | Mean | SD | Min | Max | Skewness | Kurtosis |
|---|---|---|---|---|---|---|---|---|
| All R&D cooperation | 15 | 611 | 40.7333 | 137.27 | 1 | 536 | 3.45 | 12.97 |
| Cross-city R&D cooperation | 11 | 62 | 5.6364 | 9.97 | 1 | 33 | 2.18 | 6.35 |
| Intra-provincial cross-city R&D cooperation | 4 | 5 | 1.25 | 0.5 | 1 | 2 | 1.15 | 2.33 |
| Inter-provincial cross-city R&D cooperation | 7 | 57 | 8.1428 | 12.06 | 1 | 33 | 1.46 | 3.61 |

Tianjin-Hebei urban agglomeration was proposed in February 2014, compared with the data of Yangtze River Delta urban agglomeration, the amount of R&D cooperation within the Beijing-Tianjin-Hebei urban agglomeration was far lower. According to the data of the three time periods, the overall R&D cooperation volume was increasing. The total amount of intra-provincial cross-city R&D cooperation was less than that of inter-provincial R&D cooperation, while the average cooperation intensity was higher than that of inter-provincial R&D cooperation.

**3.1.3 Pearl River Delta urban agglomeration.** Tables 8–10 lists the results of a descriptive statistical analysis of R&D cooperation in cities from the Pearl River Delta urban agglomeration. Since the special administrative regions of Hong Kong and Macao within the Pearl River Delta urban agglomeration were not considered, cities in the Pearl River Delta are all located in Guangdong province, so the amount of cross-city R&D cooperation should be equal to that of intra-provincial cross-city R&D cooperation, without inter-provincial R&D cooperation considered. According to the data of the three time periods in the table, the trend of overall R&D cooperation volume of the Pearl River Delta was consistent with those of the Yangtze River Delta and the Beijing-Tianjin-Hebei urban agglomerations, which was also rising. In addition, the amount of R&D cooperation increased significantly from 2014 to 2016, however, the amount of cross-city R&D cooperation did not increase significantly, presumably because the policy advantages of the special zone contributed to close cooperation between Guangzhou and Shenzhen.

**Table 8. Descriptive statistics of the Pearl River Delta 2008–2010.**

|  | ME | Sum | Mean | SD | Min | Max | Skewness | Kurtosis |
|---|---|---|---|---|---|---|---|---|
| All R&D cooperation | 8 | 107 | 13.375 | 20.06 | 1 | 60 | 1.81 | 4.84 |
| Cross-city R&D cooperation | 4 | 13 | 3.25 | 3.30 | 1 | 8 | 0.90 | 2.09 |
| Intra-provincial cross-city R&D cooperation | 4 | 13 | 3.25 | 3.30 | 1 | 8 | 0.90 | 2.09 |

**Table 9. Descriptive statistics of the Pearl River Delta 2011–2013.**

|  | ME | Sum | Mean | SD | Min | Max | Skewness | Kurtosis |
|---|---|---|---|---|---|---|---|---|
| All R&D cooperation | 17 | 218 | 12.8235 | 23.24 | 1 | 89 | 2.39 | 7.96 |
| Cross-city R&D cooperation | 11 | 60 | 5.4545 | 10.31 | 1 | 36 | 2.68 | 8.50 |
| Intra-provincial cross-city R&D cooperation | 11 | 60 | 5.4545 | 10.31 | 1 | 36 | 2.68 | 8.50 |

**Table 10. Descriptive statistics of the Pearl River Delta 2014–2016.**

|  | ME | Sum | Mean | SD | Min | Max | Skewness | Kurtosis |
|---|---|---|---|---|---|---|---|---|
| All R&D cooperation | 23 | 1221 | 53.0869 | 145.71 | 1 | 659 | 3.53 | 14.60 |
| Cross-city R&D cooperation | 14 | 164 | 11.7143 | 16.46 | 1 | 61 | 2.09 | 6.81 |
| Intra-provincial cross-city R&D cooperation | 14 | 164 | 11.7143 | 16.46 | 1 | 61 | 2.09 | 6.81 |

**Table 11. Top 10 cities, cross-city cooperation, based on common patent data of the Yangtze River Delta Urban agglomeration, 2008–2016.**

| Period | 2008–2010 | | 2011–2013 | | 2014–2016 | | 2008–2016 | |
|---|---|---|---|---|---|---|---|---|
| Rank | City pair | Number | City pair | Number | City pair | Number | City pair | Number |
| 1 | Nanjing and Maanshan | 32 | Shanghai and Suzhou | 41 | Nanjing and Suzhou | 63 | Shanghai and Suzhou | 111 |
| 2 | Shanghai and Suzhou | 29 | Nanjing and Zhenjiang | 26 | Nanjing and Zhenjiang | 49 | Nanjing and Suzhou | 86 |
| 3 | Nanjing and Taizhou1 | 28 | Shanghai and Changzhou | 25 | Shanghai and Hangzhou | 48 | Nanjing and Zhenjiang | 77 |
| 4 | Shanghai and Nanjing | 18 | Shanghai and Nanjing | 24 | Shanghai and Suzhou | 41 | Shanghai and Hangzhou | 72 |
| 5 | Changzhou and Yancheng | 14 | Nanjing and Suzhou | 22 | Suzhou and Zhenjiang | 28 | Shanghai and Nanjing | 54 |
| 6 | Shanghai and Hefei | 8 | Shanghai and Wuxi | 18 | Shanghai and Taizhou1 | 21 | Nanjing and Taizhou1 | 44 |
| 7 | Shanghai and Taizhou2 | 8 | Shanghai and Hangzhou | 18 | Shanghai and Wuxi | 17 | Shanghai and Changzhou | 43 |
| 8 | Shanghai and Jiaxing | 8 | Shanghai and Shaoxing | 16 | Shanghai and Shaoxing | 16 | Shanghai and Wuxi | 40 |
| 9 | Shanghai and Jinhua | 7 | Hangzhou and Shaoxing | 16 | Shanghai and Nantong | 12 | Shanghai and Shaoxing | 38 |
| 10 | Shanghai and Hangzhou | 6 | Shanghai and Zhenjiang | 11 | Shanghai and Nanjing | 12 | Suzhou and Zhenjiang | 35 |

## 3.2 The top 5 cities, cross-city cooperation

Tables 11–13 shows the city pairs with high frequency of cross-city cooperation within the three major urban agglomerations.

For the Yangtze River Delta urban agglomeration, 10 pairs of cities with high frequency of cross-city cooperation were selected. From 2008 to 2010, the proportion of one party being Shanghai was (7/10), and the proportion of one party being a municipality directly under the central government or provincial capital city was (9/10). From 2011 to 2013, the proportion of one party being Shanghai was (7/10), and the proportion of one party being a municipality directly under the central government or provincial capital city was (20/20); From 2012 to

**Table 12. Top 5 cities, cross-city cooperation, based on common patent data of the Beijing-Tianjin-Hebei Urban agglomeration, 2008–2016.**

| Period | 2008–2010 | | 2011–2013 | | 2014–2016 | | 2008–2016 | |
|---|---|---|---|---|---|---|---|---|
| Rank | City pair | Number | City pair | Number | City pair | Number | City pair | Number |
| 1 | Beijing and Tianjin | 20 | Beijing and Tianjin | 20 | Beijing and Tianjin | 33 | Beijing and Tianjin | 83 |
| 2 | Beijing and Shijiazhuang | 9 | Beijing and Shijiazhuang | 8 | Beijing and Shijiazhuang | 15 | Beijing and Shijiazhuang | 32 |
| 3 | Shijiazhuang and Handan | 2 | Beijing and Chengde | 2 | Beijing and Qinhuangdao | 4 | Beijing and Qinhuangdao | 5 |
| 4 | Beijing and Baoding | 1 | Shijiazhuang and Qinhuangdao | 2 | Shijiazhuang and Qinhuangdao | 2 | Shijiazhuang and Qinhuangdao | 4 |
| 5 | ----- | -- | Beijing and Baoding | 1 | Beijing and Langtang | 2 | Beijing and Chengde | 3 |

**Table 13. Top 5 cities, cross-city cooperation, based on common patent data of the Pearl River Delta Urban agglomeration, 2008–2016.**

| Period | 2008–2010 | | 2011–2013 | | 2014–2016 | | 2008–2016 | |
|---|---|---|---|---|---|---|---|---|
| Rank | City pair | Number | City pair | Number | City pair | Number | City pair | Number |
| 1 | Guangzhou and Foshan | 8 | Guangzhou and Dongguan | 36 | Guangzhou and Dongguan | 61 | Guangzhou and Dongguan | 100 |
| 2 | Guangzhou and Dongguan | 3 | Guangzhou and Foshan | 7 | Guangzhou and Shenzhen | 27 | Guangzhou and Foshan | 36 |
| 3 | Guangzhou and Shenzhen | 1 | Guangzhou and Zhuhai | 4 | Guangzhou and Foshan | 21 | Guangzhou and Shenzhen | 32 |
| 4 | Zhongshan and Zhaoqing | 1 | Guangzhou and Shenzhen | 4 | Guangzhou and Zhuhai | 17 | Guangzhou and Zhuhai | 21 |
| 5 | ----- | -- | Guangzhou and Zhongshan | 2 | Guangzhou and Zhaoqing | 12 | Guangzhou and Zhaoqing | 13 |

2014, the proportion of one party being Shanghai was (7/10), and the proportion of one party being a municipality directly under the central government or provincial capital city was (9/10). From city pairs with high frequency of cross-city cooperation, it can be preliminarily seen that the cross-city cooperation between Shanghai and Jiangsu province was far more than that between Zhejiang province and Anhui province because of the solid foundation of biomedical industries in Jiangsu province, such as Hengrui Medicine, Yangtze River Pharmaceutical Group and Taizhou pharmaceutical high-tech industrial park. In terms of comprehensive innovation capability, Shanghai and Jiangsu province led the forefront. There was a large difference in the comprehensive innovation capability among cities in Zhejiang Province. Hangzhou was the main engine of innovation development in Zhejiang province. Cities like Hefei and Wuhu were accelerating the catch-up under the advantages of scientific-technological innovation.

For the Beijing-Tianjin-Hebei urban agglomeration, five pairs of cities with high frequency of cross-city cooperation were selected. From 2008 to 2010, the proportion of one party being Beijing was (3/4), and the proportion of one party being a municipality directly under the central government or a provincial capital city was (4/4). From 2011 to 2013, the proportion of one party being Beijing was (4/5), and the proportion of one party being a municipality directly under the central government or provincial capital city was (5/5); From 2012 to 2014, the proportion of one party being Beijing was (4/5), and the proportion of one party being a municipality directly under the central government or provincial capital city was (5/5). From city pairs with high frequency of cross-city cooperation, it can be preliminarily seen that the cross-city cooperation within Beijing-Tianjin-Hebei urban agglomeration was mainly concentrated within the range of municipalities directly under the central government and provincial capital city with good economic development, also indirectly reflecting the characteristic of imbalanced economic development in China.

For the Pearl River Delta urban agglomeration, five pairs of cities with high frequency of cross-city cooperation were selected. From 2008 to 2010, the proportion of one party being Guangzhou was (3/4), and the proportion of one party being a municipality directly under the central government or a provincial capital city was (3/4). From 2011 to 2013, the proportion of one party being Guangzhou was (5/5), and the proportion of one party being a municipality directly under the central government or provincial capital city was (5/5); From 2012 to 2014, the proportion of one party being Guangzhou was (5/5), and the proportion of one party being a municipality directly under the central government or a provincial capital city was (5/5). From city pairs with high frequency of cross-city cooperation, it can be preliminarily seen that most cross-city cooperation within the Pearl River Delta urban agglomeration was concentrated in Guangzhou, a first-tier city with rapid economic development, as well as reflected in the spatial interaction model below.

### 3.3 Results of spatial interaction model

Table 14 is the estimation result of the negative binomial spatial interaction model of common patent data among cities of Yangtze River Delta urban agglomeration. Due to the small amount of data of Beijing-Tianjin-Hebei urban agglomeration and Pearl River Delta urban agglomeration, the spatial interaction model of them was not divided into three time periods. Table 15 is the estimation result of the negative binomial spatial interaction model of common patent data of Beijing-Tianjin-Hebei urban agglomeration and Pearl River Delta urban agglomeration. The spatial interaction model showed that the discrete coefficient was significant to the negative binomial model and the overdispersion test rejected the null hypothesis: The conditional variance of the dependent variable was equal to the conditional mean of the

**Table 14. Estimation results of the negative binomial spatial interaction model, based on common patents among cities in Yangtze River Delta urban agglomeration.**

| Years | 2008–2010 | | 2011–2013 | | 2014–2016 | |
|---|---|---|---|---|---|---|
| Model variables | Coefficient beta | Standard errors | Coefficient beta | Standard errors | Coefficient beta | Standard errors |
| **Scale effects** | | | | | | |
| Original variable [$\alpha_1$] | -0.8478 *** | 0.3242 | 0.0896 *** | 0.4073 | 0.2744 *** | 0.4197 |
| Target variable [$\alpha_2$] | -0.0655 *** | 0.2459 | 0.3164*** | 0.2741 | 0.8402*** | 0.3096 |
| **Spatial factors** | | | | | | |
| Geographical distance [$\beta_1$] | -0.2975*** | 0.2991 | -0.4066*** | 0.3121 | -0.3164*** | 0.3166 |
| High-speed railway [$\beta_2$] | 0.2298 *** | 0.3313 | 0.5655*** | 0.3277 | 0.3501*** | 0.3878 |
| **Economic and technological factors** | | | | | | |
| Economic gap[$\beta_3$] | 0.1407*** | 0.1487 | -0.0113*** | 0.1958 | 0.4938*** | 0.2672 |
| First-tier city dummy variable[$\beta_4$] | 1.4312*** | 0.7503 | 0.7006*** | 0.4573 | -0.1519*** | 0.6503 |
| **Political bias factors** | | | | | | |
| Different province dummy variable[$\beta_5$] | 0.3943*** | 0.4633 | -0.3448*** | 0.4548 | -0.9702*** | 0.4838 |
| Provincial capital city dummy variable[$\beta_6$] | 0.2491*** | 0.4186 | 0.9229*** | 0.3872 | 0.5060*** | 0.3897 |
| Central city dummy variable[$\beta_7$] | ---- | ---- | 1.8195*** | 0.6385 | 1.4354*** | 0.6526 |
| **R & D environment factor** | | | | | | |
| Industry dummy variable [$\beta_8$] | -0.2296*** | 0.1852 | -0.0741 | 0.1817 | -0.0436 | 0.1487 |

*** Statistically significant at the 0.001 significance level; ** Statistically significant at the 0.01 significance level.

dependent variable. Therefore, the spatial interaction model adopted negative binomial regression, indicating that the parameter estimation was highly statistically significant in the negative binomial interaction model.

**3.3.1 Yangtze River Delta urban agglomeration.** 2008–2010: $\alpha_1$ = -0.8478, $\alpha_2$ = -0.0655; 2011–2013: $\alpha_1$ = 0.0896, $\alpha_2$ = 0.3164; 2010–2014: $\alpha_1$ = 0.2744, $\alpha_2$ = 0.8402. The scale variable parameter indicated that during 2008–2010, the greater number of enterprises in a city, the

**Table 15. Estimation results of the negative binomial spatial interaction model, based on common patents among cities in Beijing-Tianjin-Hebei and Pearl River Delta urban agglomerations (2018–2014).**

| Years | Beijing-Tianjin-Hebei | | Pearl River Delta | |
|---|---|---|---|---|
| Model variables | Coefficient beta | Standard errors | Coefficient beta | Standard errors |
| **Scale effects** | | | | |
| Original variable [$\alpha_1$] | -2.9581 *** | 1.0422 | 0.0225 *** | 0.9089 |
| Target variable [$\alpha_2$] | 0.3544 *** | 0.3868 | 0.6556*** | 0.3683 |
| **Spatial factors** | | | | |
| Geographical distance [$\beta_1$] | -0.0814*** | 0.3841 | -0.2845*** | 0.107 |
| High-speed railway [$\beta_2$] | 0.2156*** | 0.4944 | 1.0593*** | 0.4261 |
| **Economic and technological factors** | | | | |
| Economic gap[$\beta_3$] | 0.8570*** | 1.0642 | -0.2573*** | 0.2938 |
| First-tier city dummy variable[$\beta_4$] | -0.7386*** | 2.5269 | 0.9628*** | 1.2990 |
| **Political bias factors** | | | | |
| Provincial capital city dummy variable[$\beta_6$] | -1.0129*** | 0.9834 | 0.7963*** | 0.5603 |
| Central city dummy variable [$\beta_7$] | ---- | ---- | 0.7963*** | 0.5603 |
| **R & D environment factor** | | | | |
| Industry dummy variable [$\beta_8$] | -2.1596 | 0.8614 | 1.1371*** | 0.4034 |

*** Statistically significant at the 0.001 significance level; ** Statistically significant at the 0.01 significance level.

less possibility of cooperation between the city and other cities. However, from 2011 to 2013 and 2014 to 2016, the more enterprises, the more possibility of cross-city cooperation, which might be related to the situation of economy and policies of the city during each period. The regression coefficient of the spatial variable parameter β1 demonstrated that geographical distance between cities had a significant negative influence on the possibility of cooperation. The regression coefficient β2 indicated that the possibility of cooperation would increase for cities with high-speed railway connection. However, contrary to expectations, the authors found that economic differences between cities had a positive effect on the probability of cooperation (2008–2010:β3 = 0.1407; 2014–2016:β3 = 0.4938); Dummy variable regression of first-tier cities (2008–2010:β4 = 0.1407; 2011–2013: β4 = -0.7006; 2014–2016:β4 = -0.1519) demonstrated negative influence on the cooperation between two cities, and the advantages of first-tier cities were getting smaller and smaller.

Regression coefficient β5 indicated that cooperation possibility would decrease when two cities were in different provinces. In addition, regression coefficients β6 and β7 indicated that cooperation possibility between two cities might increase when one or both of them were provincial capitals and the other was the center of urban agglomeration. Results of variable parameters of R&D factor (2008–2010: β8 = -0.2296; 2011–2013, β8 = -0.0741; 2014–2016: β8 = -0.0435) verified our previous conjecture: The smaller development difference of pharmaceutical industry between the two cities, the more R&D cooperation between the two cities.

**3.3.2 Beijing-Tianjin-Hebei urban agglomeration.** The trends of spatial interaction model results of Beijing-Tianjin-Hebei urban agglomeration were almost the same as those of Yangtze River Delta urban agglomeration. In spatial factors, the geographical distance variable β1 was -0.0814, proving geographical distance between cities had a significant negative impact on the possibility of cooperation. The dummy variable of high-speed railway β2 was 0.2156, indicating that the possibility of cooperation between two cities would be increased after the high-speed railway connection is available. The economic difference between two cities β3 was 0.8670, showing a positive effect on the probability of cooperation. The regression of dummy variables of first-tier cities β4 was -0.7386, demonstrating negative influence on the cooperation between two cities. In addition, since Beijing and Tianjin were not considered as provincial capital cities in the statistics, β6 was -1.0129, indicating that if one of the cooperative cities was Beijing or Tianjin, it would promote cooperation more than Shijiazhuang(the provincial capital). The regression coefficient showed that when one or both cities were the center of urban agglomeration, the cooperation possibility between the two cities might increase. R&D factor variable parameter β8 was -2.1596, indicating that the smaller development difference between the two cities, the more R &D cooperation between the two cities.

**3.3.3 Pearl River Delta urban agglomeration.** The results of spatial interaction model of the Pearl River Delta urban agglomeration were quite different from those of Yangtze River Delta urban agglomeration and Beijing-Tianjin-Hebei urban agglomeration. In spatial factors, the geographical distance variable β1 was -0.2845, which proved that the geographical distance between cities had a significant negative impact on the possibility of cooperation. The dummy variable of high-speed railway β2 was 1.0593, demonstrating that the possibility of cooperation between two cities would be increased after the high-speed railway connection was available. The regression coefficients β6 and β7 indicated that cooperation possibility between two cities might increase if one or both of them were provincial capitals and one was the center of urban agglomeration. These results were consistent with those of the other two major urban agglomerations. However, results of variables of economic difference and dummy variables of first-tier cities were different from those of the two major urban agglomerations, which could support the author's previous conjecture. β3 = -0.2573, indicating that the greater economic difference between the two cities, the more cooperation decreases. If there was a first-tier city

between the two cities, the cooperation possibility between the two cities will be increased. However, the R&D factor variable result ($\beta_8$ = 1.1371) indicated otherwise compared to the author's conjecture, that the greater development difference between the two cities, the more R&D cooperation between the two cities.

## 3.4 Spatial structure evolution

Do three times across the city development cooperation space network evolution: the three urban agglomerations of biological medicine industry cooperation across the city in 2008–2016 frequency matrix using R language mapping the three urban agglomerations of biological medicine industry innovation cooperation spatial structure evolution diagram is shown in Fig 1. We are going to the Yangtze river delta urban agglomeration was divided into 26 region, the pearl river delta urban agglomeration and beijing-tianjin-hebei urban agglomeration, respectively divided into 15 and 14 regions, each region represents a prefecture level, is our city the study sample. Regional connection line to represent the I and j city, contact cooperation. With the extent of the straight line thickness and color display, I and j city r&d cooperation frequency, the degree of thickness, the color of the line contact with the number of times a consistent, line is thicker, the deeper the color, the higher the number of the representative city cooperation.

Overall, the development of the Yangtze river delta urban agglomeration between cities is relatively close, crisscrossed innovation cooperation network can let each city for knowledge

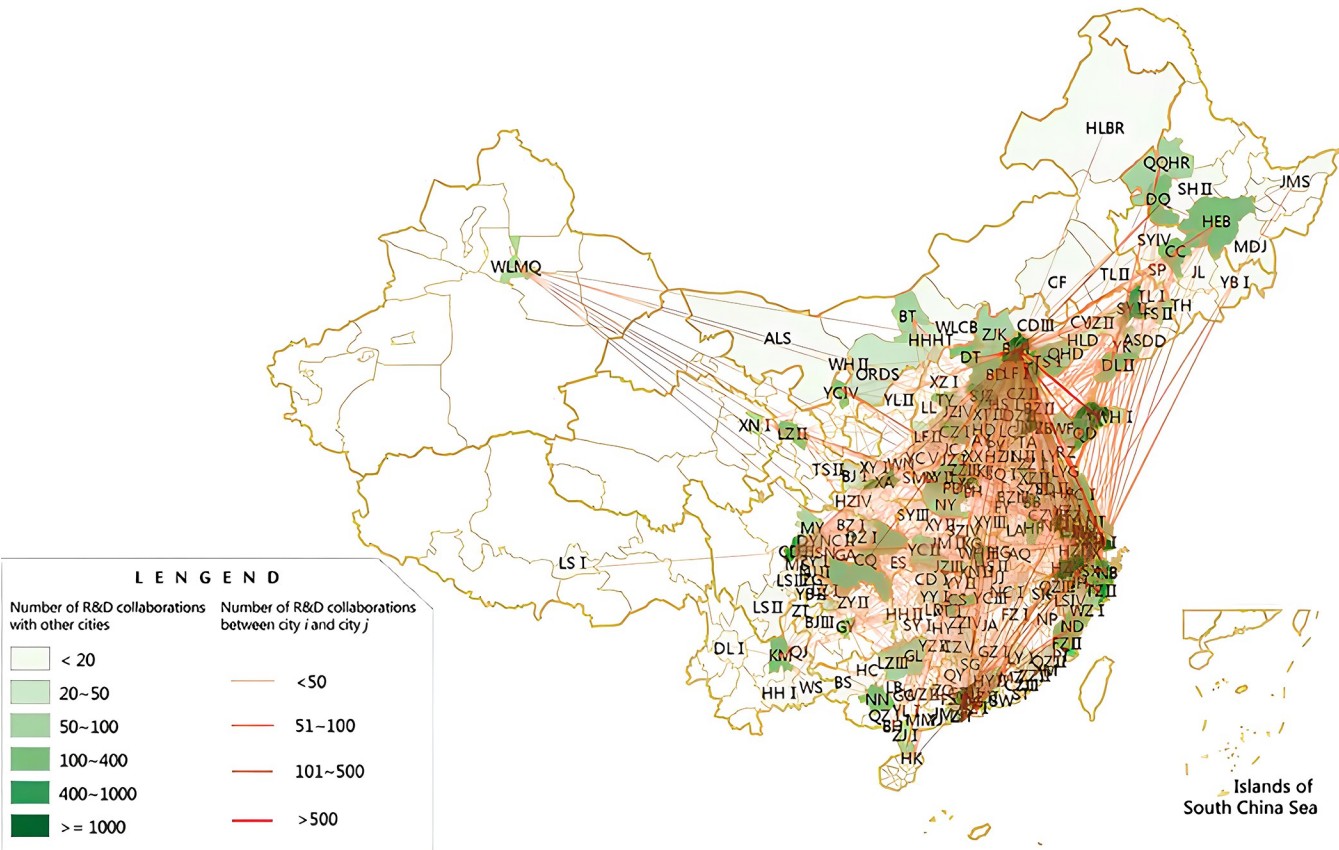

**Fig 1. Three bay area in 2008–2016 urban agglomeration of biological medicine industry innovation cooperation spatial structure diagram.**

transfer and sharing of resources, realize the innovation of the urban agglomeration highway cooperation. Between the pearl river delta urban agglomeration from city to city innovation cooperation network become freely crisscross, but only in some cities, geo-graphic distance far away between guangzhou, shenzhen and other cities and other cities still not have a high level of knowledge transfer and sharing resources, urban agglomeration innovation cooperation tightness is not strong. Innovation cooperation network of the beijing-tianjin-hebei urban agglomeration between cities become freely crisscross, but part of the city or not into innova-tion cooperation network, most of the city to speed up the innovation cooperation between knowledge transfer and sharing of resources, speed up the innovation cooperation of urban agglomerations. (in addition to the individual cities, the development of beijing-tianjin-hebei urban agglomeration between cities, crisscrossed innovation cooperation network can let between cities for knowledge transfer and sharing of resources, realize the innovation of the urban agglomeration highway cooperation).

## 4. Discussion

In this paper, for the first time, multiple cities were used as spatial analysis units to study the main influencing factors of inter-regional biomedical R&D cooperation in China, with the. purpose of analyzing the variables of cross-city R&D cooperation by using cooperative patent data. Due to excessive discreteness of the data, the author adopted the negative binomial regression model to analyze the influencing factors of biomedical R&D cooperation in Yangtze River Delta urban agglomeration during three time periods: 2008–2010, 2011–2013, 2014–2016,as well as the influencing factors of biomedical R&D cooperation in Beijing-Tianjin-Hebei urban agglomeration and Pearl River Delta urban agglomeration from 2008 to 2016, test results showed that the independent variables of the negative binomial regression model of the independent variables were significant. And do three times across the city development cooperation space network evolution figure more image analysis spatial pattern characteristics of biological medicine research and development cooperation.

### 4.1 Yangtze River Delta urban agglomeration

Previously, it was speculated that the possibility of cross-city R&D cooperation would decrease significantly with the increasing geographical distance and increase significantly with the avail-ability of high-speed railway transportation. These two findings were consistent with previous empirical studies [23, 24].

In the regression model of Yangtze River Delta urban agglomeration, regression coefficients (economic difference) of 2008–2010 and 2014–2016 were $\beta 3 = 0.1407$, $\beta 3 = 0.4938$ respectively, suggesting the economic difference between any two cities had a significant positive impact on the frequency of cross-city biomedical R&D cooperation, which was contrary to our literature review results suggesting that the economic difference between two regions had a significant negative effect on their collaboration probability [12, 25, 26], while the regression coefficient (economic difference) of 2011–2013 was $\beta 3 = -0.0113$. The author speculated that the cities of R&D cooperation within Yangtze River Delta urban agglomeration were more like large cities, such as Shanghai and Nanjing. Although the economic development levels of Shanghai, Nan-jing, and other large cities were quite different, on basisi of economic and human resource advantages of Shanghai, as well as technology and policy advantages of Nanjing, Suzhou and Taizhou, developed cities could still attract disadvantaged cities to complement each other's advantages and resources.

Regression coefficients of first-tier cities were 2008–2010: $\beta 4 = 1.4319$; 2011–2013:$\beta 4 = -0.7006$; 2014–2016:$\beta 4 = -0.1519$. Regression coefficients of different provinces were 2008–

2010:β5 = 0.3943; 2011–2013:β5 = -0.3448; 2014–2016: β5 = -0.9702. The reasons for such results might be that the development planning of Yangtze River Delta was based on the urbanization of the core area, which would first promote the urbanization of adjacent cities, and then gradually promote regional economic integration. The core area was centered in Shanghai and included other mutually reinforcing economic areas such as Hangzhou, Shaoxing, and Ningbo in Zhejiang province as well as Suzhou, Wuxi, and Changzhou in Jiangsu province. Therefore, with the prolongation of time, the influence of first-tier cities would begin to decline, and the cooperation within the same province would increase, that regional cooperation would be more likely to occur when two cities were located in the same province. It also suggested that inter-provincial, cross-city R&D cooperation might encounter significant obstacles from certain political factors. These political factors could be considered as "spatial province bias". It could be seen from the comparison of the average intensity of intra-provincial cooperation, that the results reflected a spatial province bias, which might be caused by local protectionism. In China, 93% of public research institutions are mainly invested and established by local governments. The vast majority of R&D funds belong to government policymakers in China [27]. China's R&D system is dominated by top-down institutions, while bottom-up research organizations only account for a small proportion [28]. Therefore, the policy advantage of local government would be relatively biased toward the research organizations in the province.

Through descriptive analysis and econometric model analysis, the superior position of provincial capitals and central cities in the cross-city biomedical R&D cooperation network of Yangtze River Delta was also verified. There are more scientific research institutions in provincial capitals and urban agglomeration centers, and they have more rights to obtain R&D cooperation opportunities. The results summarize that the biomedical R&D cooperation network is consistent with the urban cooperation network, and the provincial political center cities play an important role in R&D cooperation.

## 4.2 Beijing-Tianjin-Hebei urban agglomeration

The regression coefficient of geographical distance was β1 = -0.0814, which was not significant compared with previous results (the regression coefficient ranges from -0.228 to -0.354) [10–12, 23, 24]. A possible reason is that, within the entire cooperation network, the geographic distance between cooperative cities in the same province is usually short, while the geographic distance between cooperative cities in two provinces is relatively long. There is a certain degree of overlap between dummy variables and geographic distance variables of different provinces and the influence on cooperation frequency exceeds the that from geographic distance variables, which leads the geographical distance variables to be insignificant in the model and fails to pass statistical tests. The results demonstrated that the provincial administrative bias factor had a greater impact on cross-city biomedical R&D cooperation.

In the regression model of Beijing-Tianjin-Hebei urban agglomeration, the economic difference between any two cities had a significant positive impact on the frequency of cross-city biomedical R&D cooperation (β3 = 0.8570). It was contrary to the results of previous literature reviews, which suggested that economic differences might have negative impact on the possibilities of R&D cooperation between two cities. The hypothesis was consistent with the research on the positive effect of economic difference on cross-city regional R&D cooperation, which was verified by the urban R&D cooperation in China. The authors proposed several possible factors to explain the positive results. Firstly, from the perspective of descriptive statistical analysis, R&D cooperation was mostly concentrated in Beijing and Tianjin, and cross-city cooperation was associated with these two megacities. That is, Beijing and Tianjin still

attracted other cities in Hebei to cooperate with them despite the great differences in the level of economic development. Secondly, Beijing and Tianjin had superior political, economic, technological and human resource advantages. Developed cities and less developed cities could cooperate to complement each other's advantages and resources to jointly promote development. However, the regression coefficient of first-tier cities was $\beta4 = -0.7386$, which was inconsistent with our previous estimation. The authors speculated that in the early stage of the development the Beijing-Tianjin-Hebei urban agglomeration, the regional development was quite different; because the two big cities of Beijing and Tianjin were too "obese", while the surrounding small and medium-sized cities were too "thin". With the collaborative development of the Beijing-Tianjin-Hebei region, Shijiazhuang gradually rose as a new star, with a rapid growth in its innovation cooperation frequency, much faster than the level of economic growth, constantly attracting surrounding cities and driving the development of cities such as Handan, Changzhou, and Xingtai, etc. Then the inter-regional differences within Beijing-Tianjin-Hebei urban agglomeration gradually decreased, while cooperation among other cities increased. Moreover, Beijing as the only one first-tier city within the region of Beijing-Tianjin-Hebei urban agglomeration, if other cities increase the amount of cooperation, the advantage of Beijing being a first-tier city would slowly decline. In addition, in the regression model, when one party was a provincial capital, there was a negative impact on the R&D cooperation between the two cities. Because in the data analysis, Beijing and Tianjin as municipalities directly under the central government were not counted as provincial capital cities. The regression coefficient of industry variables of R&D factors was $\beta8 = -2.1596$, indicating that the smaller difference between the two cities' biomedical industries, the easier it is to promote cooperation.

## 4.3 Pearl River Delta urban agglomeration

The Pearl River Delta urban agglomeration was the same as the other two urban agglomerations, supported by the results of the two variables of spatial factors [23, 24]. The possibility of cross-city R&D cooperation would decrease significantly with the increase of geographical distance and increase significantly with high-speed railway transportation available. Moreover, the regression coefficient of high-speed rail $\beta2 = 1.0593$ greatly exceeded the expected value. Because the transportation development of the Pearl River Delta urban agglomeration was mainly concentrated in the large-scale port cities such as Guangzhou, Shenzhen, and Zhuhai. Currently, the high-speed rail between Guangzhou and Shenzhen to Foshan has not been opened, which greatly affects the cooperation and exchanges in these areas.

Hypothesis 4 verified that the existence of first-tier cities between two cities would further promote cooperation between them. Hypothesis 6 and hypothesis 7 also supported that political bias factors of cities were the main positive factors affecting cross-city R&D cooperation. The author believes that because Guangzhou is the central city of the Pearl River Delta urban agglomeration, its development has huge advantages, such as its vast hinterland is the main hub for the vast region of South China to the world and North China, and the main gateway for the world into South China; Meanwhile, as the capital of Guangdong province with strong economic strength, it has become the most important business center, comprehensive transportation hub and passenger flow gathering place in South China. Therefore, the state has given Guangzhou the absolute preference of high-speed development.

However, the influencing factor coefficient of industry from the regression model was $\beta8 = 1.1371$, inconsistent with our previous conjecture. It indicated that the greater difference in pharmaceutical industry between the two cities, the better cooperation between them. This might be because of the SAR policy changes in recent years, the Pearl River Delta must follow

the main function orientated regional spatial layout optimization, to achieve the situation of integrated and coordinated development. All cities were required to develop in a dislocated manner to form a complementary and mutually reinforcing development pattern. Furthermore, central cities such as Guangzhou and Shenzhen should make full use of their advantages in economy, human resource, technologies and policies to cooperate with Zhuhai, Zhongshan, and other cities with complementary advantages and resources.

### 4.4 Limitations of the study

The results of the study still had some limitations. Since the influencing factors of R&D cooperation are more complex in other studies, the spatial interaction model of this study can explain more variations or changes of the influencing factors, such as technology proximity. Therefore, the existing study can be improved by introducing other new independent variables.

## 5 Conclusion

### 5.1 Yangtze River Delta urban agglomeration

In the Yangtze River Delta urban agglomeration, the geographic distance was negatively correlated with the frequency of cross-city biomedical R&D cooperation. Cities with high-speed railway connection were positively correlated with cross-city biomedical R&D cooperation. The economic differences between cities had a positive impact on the frequency of biomedical R&D cooperation, that small cities with slower economic development would actively seek cooperation with core cities to obtain resources. It indicated that urban location might be a significant factor affecting the cross-city biomedical R&D cooperation, and the imbalance of regional economic development had a significant, sustained, and inevitable impact on it. In addition, the influence of administrative bias on biomedical R&D cooperation still existed, which also verified the previous factors affecting urban R&D cooperation. Finally, the smaller development difference of biomedical industry between two cities, the more beneficial cooperation between them.

### 5.2 Beijing-Tianjin-Hebei urban agglomeration

In the Beijing-Tianjin-Hebei urban agglomeration, the geographical distance was negatively correlated with the frequency of cross-city biomedical R&D cooperation; High-speed railway transportation had a positive correlation with the frequency of cross-city biomedical R&D cooperation. The economic differences between cities were positively correlated with the frequency of biomedical R&D cooperation, which was completely consistent with the situation of Yangtze River Delta urban agglomeration. In addition, the influence of administrative bias on biomedical R&D cooperation still existed, but the conclusion was inconsistent with that of Yangtze River Delta urban agglomeration. The existence of first-tier cities did not promote the development between the two cities. Finally, the smaller development difference of biomedical industry between two cities, the more beneficial cooperation between them.

### 5.3 Pearl River Delta urban agglomeration

The Pearl River Delta urban agglomeration was consistent with the first two urban agglomerations. The geographical distance was negatively correlated with the frequency of cross-city biomedical R&D cooperation, while high-speed railway transportation was positively correlated with the cross-city biomedical R&D cooperation. In addition, economic differences between cities could negatively influence the frequency of biomedical R&D cooperation. The greater development difference of biomedical industry between two cities, the more beneficial cooperation

between them, which was different from the situation of the other two urban agglomerations in the study. Finally, the influence of administrative bias on biomedical R&D cooperation still existed, which also verified the influencing factors of previous urban R&D cooperation.

## 5.4 Summary

Although the development strategies of the three urban agglomerations studied in this paper are different based on their actual situations, the routes they take are generally the same. Each urban agglomeration needs a central city, and only by surrounding this "central city" can an urban agglomeration be formed. Therefore, the development strategies of the three urban agglomerations are to vigorously develop the central city respectively within the urban agglomeration and form the core leading city. Then, to strengthen the development of cities with small economic differences from the leading city, and gradually form a city circle with the core leading city as the center. Finally, within their respective urban areas, large cities drive small cities to promote the transfer of pharmaceutical industry, forming multi-level industrial clusters with gradient development and reasonable division of labor, and establishing an industrial cooperation system with complementary advantages and mutual benefit.

The Yangtze River Delta region should continue to develop the network spatial pattern of "one core, five circles, and four belts". "One core" refers to Shanghai, the economic center of China, which plays a leading role in promoting and driving the development of regional central cities. "Five circles" refer to the Nanjing metropolitan circle, Hangzhou metropolitan circle, Hefei metropolitan circle, Suzhou-Wuxi-Changzhou metropolitan circle, and Ningbo metropolitan circle. "Four belts" refer to development belts of Shanghai-Nanjing-Hangzhou-Ningbo. So that a scalable system of central cities (megacities)-large cities-medium cities -small cities could be established, forming a biomedical industry with multiple forms, wide fields and in-depth development.

For the Beijing-Tianjin-Hebei region, two megacities have led the development of the regional urban agglomeration. As the national political center, cultural center, international exchange center, and scientific and technological innovation center, Beijing has built a "high-quality, high-end and sophisticated industrial structure". Tianjin is practicing high-quality green development policy and will vigorously develop the biomedical industry. However, due to the huge economic differences among cities except for Beijing and Tianjin, the main purpose of Beijing-Tianjin-Hebei region is to narrow the differences between different cities, achieve common development, establish a more close "relationship" between each other and realize win-win cooperation.

For the Pearl River Delta region, it can continue to take Guangzhou and Shenzhen as core cities to drive other regions and build Guangzhou into an international metropolis facing the world and serving the whole country. Shenzhen may continue to take the advantage of being a special economic zone to become China's central port. In addition, we should make full use of the radiation service and driving role of the Pearl River Delta region. In addition to Guangzhou and Shenzhen, we should strongly support the development of pharmaceutical industry in cities like Zhuhai, Foshan, and others, and drive the east, west and north of Guangdong together with the surrounding provinces and regions as well as other regions around the Pearl River Delta region to form a coordinated development of biomedical industry.

## Supporting information

**S1 Data.**
(ZIP)

## Acknowledgments

We thank all workers for participating in the study.

## Author Contributions

**Conceptualization:** Guojun Sun, Shaoya Zhang.

**Data curation:** Shaoya Zhang.

**Formal analysis:** Shaoya Zhang.

**Funding acquisition:** Guojun Sun, Dong Zuo-jun.

**Investigation:** Shaoya Zhang.

**Methodology:** Shaoya Zhang.

**Project administration:** Shaoya Zhang, Dong Zuo-jun.

**Resources:** Shaoya Zhang.

**Software:** Shaoya Zhang, Lan Xu.

**Supervision:** Dong Zuo-jun.

**Validation:** Shaoya Zhang, Xiaoying Zhou, Shuaijun Wu.

**Visualization:** Shaoya Zhang.

**Writing – original draft:** Shaoya Zhang.

**Writing – review & editing:** Shaoya Zhang.

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
