## [Decision Letter · Decision Letter 0]

7 Jul 2022

PONE-D-22-12659The Influencing Factors of Biomedical R&D Cooperation in Three Major Urban Agglomerations of China Based on Cooperative PatentPLOS ONE

Dear Dr. Zuo-jun,

Thank you for submitting your manuscript to PLOS ONE. After careful consideration, we feel that it has merit but does not fully meet PLOS ONE’s publication criteria as it currently stands. Therefore, we invite you to submit a revised version of the manuscript that addresses the points raised during the review process.

We look forward to receiving your revised manuscript.

Kind regards,

Carlos Alberto Zúniga-González, Ph.D

Academic Editor

PLOS ONE

Journal Requirements:

    "This research was funded by Research on the Construction Path of New Drug Creation and Innovation Consortium in the Field of Life and Health in Zhejiang Province, project number: 2022C25007."

Additional Editor Comments:

Dear author your contribution is attractive and has value for impact on science, however, I consider making some improvements. Regarding the 5 points of reviewer 1, I would like to see your mean improvements. Regarding the 3 points of reviewer 2 consider that you can to improvements and make the changes. I recommend you to extend the references to 40 at least. I hope soon your changes.

Reviewers' comments:

Reviewer's Responses to Questions

**Comments to the Author**

1. Is the manuscript technically sound, and do the data support the conclusions?

Reviewer #1: Yes

Reviewer #2: Yes

2. Has the statistical analysis been performed appropriately and rigorously? 

Reviewer #1: Yes

Reviewer #2: Yes

3. Have the authors made all data underlying the findings in their manuscript fully available?

Reviewer #1: Yes

Reviewer #2: Yes

4. Is the manuscript presented in an intelligible fashion and written in standard English?

Reviewer #1: No

Reviewer #2: Yes

5. Review Comments to the Author

Reviewer #1: Using the cooperative patent data, this paper studies the influencing factors of biomedical R&D cooperation in China's Yangtze River Delta urban agglomeration, Pearl River Delta urban agglomeration and Beijing Tianjin Hebei urban agglomeration from 2008 to 2016. This paper has certain significance for promoting the scientific and technological progress of China's biomedical industry.

1. In chapter 1.1, the literature review is relative insufficient, and most reference cited in this paper was published in 2019 and before. Is it possible to consider updating the latest literature?

2. In chapter 2.1.2, research literature on influencing factors of biomedical R & D cooperation should be added.

3. In chapter 2.3.1, the reason for choosing cities in each urban agglomeration should be explained.

4. In chapter 2.3.2, the research period is divided into 2008-2010, 2011-2013 and 2014-2016. What is the basis for the division?

5. the spatial pattern characteristics of biomedical R&D cooperation have not been well analyzed in the paper, which is inconsistent with what is stated in the abstract.

Reviewer #2: I read your paper with interest.

Find my comments below:

1. There are some mistypes in your abstract that make reading and following it a bit challenging.

2. Your conclusion segment is too long and contains redundant information from the discussion.

3. There is a page with content titled “rebuttal letter” in the end. Kindly explain this. If in fact this paper had been submitted before and you’ve been asked to review the manuscript, kindly re-upload a manuscript that in fact highlights that these suggested review changes have been made.

6. PLOS authors have the option to publish the peer review history of their article (what does this mean?). If published, this will include your full peer review and any attached files.

Reviewer #1: No

Reviewer #2: No

---

## [Author Response · Author response to Decision Letter 0]

27 Sep 2022

Dear reviewers：

On behalf of all authors，thank you very much for your suggestions to us.

First of all,, for review 1 suggestion: 1 in chapter 1.1, we update the literature; 2. The proposed increase in chapter 2.1.1 biomedical research in the impacting factors of r&d cooperation in literature. This chapter mainly eliminate the standard of literature is not a tautology research literature, study design in chapter 2.2, have expounded the literature; 3. Select cities in chapter 2.3.1 is mainly based on the population base (> 21.89 million) of a standard; 4. In chapter 2.3.2, divided into periods according to the annual statistics too little, for the convenience of the statistics, chose a statistics every 2 years; 5. Increase the figure of 3.4 spatial structure evolution, more research and development of biomedical image analysis spatial pattern characteristics of cooperation.

Then, for advice from the reviewer 2:1. The part of the basic problems, such as grammar changes; 2. The conclusion part, will discuss with repetitive parts cut.

Yours sincerely.

---

## [Decision Letter · Decision Letter 1]

25 Nov 2022

Cooperation based on the patent of China's three largest urban agglomeration the influence factors of biological medicine research and development cooperation

PONE-D-22-12659R1

Dear Dr. Dong  Zou-jun,

We’re pleased to inform you that your manuscript has been judged scientifically suitable for publication and will be formally accepted for publication once it meets all outstanding technical requirements.

Kind regards,

Carlos Alberto Zúniga-González, Ph.D

Academic Editor

PLOS ONE

Additional Editor Comments (optional):

Dear authors, I apologize for late decision. I have checked your efforts and all reviewer's observations were incorporated. My sincere congratulations.

Reviewers' comments:

Reviewer's Responses to Questions

**Comments to the Author**

1. If the authors have adequately addressed your comments raised in a previous round of review and you feel that this manuscript is now acceptable for publication, you may indicate that here to bypass the “Comments to the Author” section, enter your conflict of interest statement in the “Confidential to Editor” section, and submit your "Accept" recommendation.

Reviewer #2: All comments have been addressed

2. Is the manuscript technically sound, and do the data support the conclusions?

Reviewer #2: Yes

3. Has the statistical analysis been performed appropriately and rigorously? 

Reviewer #2: Yes

4. Have the authors made all data underlying the findings in their manuscript fully available?

Reviewer #2: Yes

5. Is the manuscript presented in an intelligible fashion and written in standard English?

Reviewer #2: Yes

6. Review Comments to the Author

Reviewer #2: Thank you to the authors for taking the time to consider the reviewer feedback and assimilate the new additions into your manuscript. I believe that has added quality and strength to your manuscript.

7. PLOS authors have the option to publish the peer review history of their article (what does this mean?). If published, this will include your full peer review and any attached files.

Reviewer #2: No

---

## [Editor Report · Acceptance letter]

8 Dec 2022

PONE-D-22-12659R1 

The Influencing Factors of Biomedical R&D Cooperation in Three Major Urban Agglomerations of China Based on Cooperative Patents 

Dear Dr. Zuo-jun:

I'm pleased to inform you that your manuscript has been deemed suitable for publication in PLOS ONE. Congratulations! Your manuscript is now with our production department. 

Kind regards, 

on behalf of

Dr. Prof. Carlos Alberto Zúniga-González 

Academic Editor

PLOS ONE